# FEDMEF: TOWARDS MEMORY-EFFICIENT FEDERATED DYNAMIC PRUNING

## ABSTRACT

Federated learning (FL) promotes decentralized training while prioritizing data confidentiality. However, its application on resource-constrained devices is challenging due to the high demand for computation and memory resources for training deep learning models. Neural network pruning techniques, such as dynamic pruning, could enhance model efficiency, but directly adopting them in FL still poses substantial challenges, including post-pruning performance degradation, high activation memory, etc. To address these challenges, we propose FedMef, a novel and memory-efficient federated dynamic pruning framework. FedMef comprises two key components. First, we introduce the budget-aware extrusion that maintains pruning efficiency while preserving post-pruning performance by salvaging crucial information from parameters marked for pruning within a given budget. Second, we propose scaled activation pruning to effectively reduce activation memory, which is particularly beneficial for deploying FL to memory-limited devices. Extensive experiments demonstrate the effectiveness of our proposed FedMef. In particular, it achieves a significant reduction of 28.5% in memory footprint compared to state-of-the-art methods while obtaining superior accuracy.

## 1 INTRODUCTION

Federated learning (FL) has emerged as an important paradigm for the training of collaborative deep learning models on decentralized devices while preserving the confidentiality of local data (McMahan et al., 2017; Li et al., 2019). In particular, cross-device FL, as outlined in (Kairouz et al., 2021), places emphasis on scenarios where FL clients predominantly consist of edge devices with resource constraints. Cross-device FL has gained significant attention in academic research and industry applications, fueling a wide range of applications, including Google Keyboard (Hard et al., 2018; Leroy et al., 2019), Apple Speech Recognition (Paulik et al., 2021), etc. Despite its success, the resource-intensive nature of training models, which includes high computational and memory costs, poses challenges for the deployment of cross-device FL on resource-constrained devices.

Neural network pruning (Janowsky, 1989; Han et al., 2015; Molchanov et al., 2019b; Singh & Alistarh, 2020) is a potential solution to improve model efficiency and reduces the high demand for resources. However, a closer inspection of some preceding work on applying neural network pruning to FL (Shao et al., 2019; Li et al., 2021a; Liu et al., 2021; Munir et al., 2021) reveals a potential pitfall: They often rely on initial training of dense models, similar to centralized pruning methodologies (Han et al., 2015; Molchanov et al., 2019b; Singh & Alistarh, 2020). These federated pruning methods are not suitable for cross-device FL because training of dense models still demands high computation and memory on resource-constrained devices.

To address these challenges, recent research has shifted to federated dynamic pruning (Jiang et al., 2022; Qiu et al., 2022; Bibikar et al., 2022; Huang et al., 2022). These frameworks derive specialized pruned models by iterative adjustment of sparse on-device models. Devices start with a randomly pruned model, followed by traditional FL training, and periodically adjust the sparse model structure via pruning and growing operations (Evci et al., 2020). Through iterative training and adjustments, devices develop specialized pruned models bypassing the need to train dense models, which reduces computational and memory demands.

However, existing federated dynamic pruning frameworks (Jiang et al., 2022; Qiu et al., 2022; Bibikar et al., 2022; Huang et al., 2022) face two issues: significant post-pruning accuracy degra-

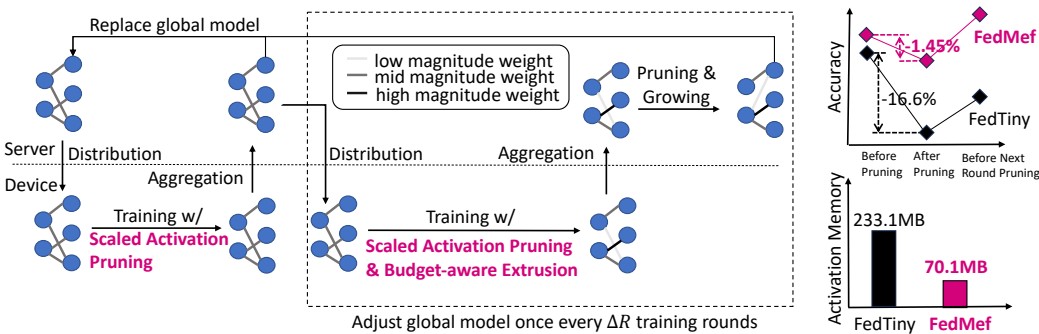

Figure 1: *Left*: Compared to baseline methods such as FedTiny (Huang et al., 2022), FedMef proposes budget-aware extrusion to preserve accuracy by transferring essential information from low-magnitude parameters to the others, and introduces scaled activation pruning to reduce memory usage. *Top right*: FedMef minimizes post-pruning accuracy loss and quickly recovers before the next round of pruning. *Bottom right*: FedMef significantly saves activation memory by more than 3 times through scaled activation pruning on the CIFAR-10 dataset with MobileNetV2.

dation and substantial activation memory usage. First, these frameworks cause a significant decline in accuracy after magnitude pruning because they hastily eliminate low-magnitude parameters, regardless of the substantial information they may contain. Such incautious parameter pruning often results in the model's inability to regain its previous accuracy before the subsequent pruning iteration, ultimately leading to suboptimal end-of-training performance. Second, these frameworks fail to reduce the memory footprint of activation. For certain widely adopted models for edge deployment, like MobileNet (Sandler et al., 2018), a significant portion of the total memory is allocated to activation memory. However, current federated dynamic pruning methods focus primarily on reducing the model size, overlooking optimization for activation memory.

In this work, we introduce FedMef, an **Fed**erated **M**emory-**ef**fcient dynamic pruning framework that adeptly addresses the aforementioned challenges. Figure 1 illustrates the workflow of FedMef and highlights our proposed two new components. First, FedMef presents *budget-aware extrusion* to address the challenge of post-pruning accuracy degradation. Rather than rashly hastily discarding low-magnitude parameters, our method salvages essential information from these potential pruning candidates by transferring it to other parameters through a surrogate loss function within a preset budget. Second, FedMef proposes *scaled activation pruning* to tackle the problem of high activation memory. This method performs activation pruning during the training process to dramatically reduce the memory footprints of the activation caches, as illustrated in Figure 2.To enhance the efficacy of scaled activation pruning, especially for devices with severe memory constraints, inspired by recent methods that eliminate batch normalization (BN) layers (Brock et al., 2021; Zhuang & Lyu, 2023), we remove conventional BN layers and replace convolution layers with Normalized Sparse Convolution (NSConv). NSConv can normalize most of the values of the activation to zero or close to it. This reduces the disparity between original and pruned activation, subsequently mitigating the degradation of accuracy during scaled activation pruning.

We conducted extensive experiments on three datasets: CIFAR-10 (Krizhevsky et al., 2009), CINIC-10 (Darlow et al., 2018), and TinyImageNet (Le & Yang, 2015), using the ResNet18 (He et al., 2016) and MobileNetV2 (Sandler et al., 2018) models. Extensive experimental results suggest that FedMef outperforms the state-of-the-art (SOTA) methods on all datasets and models. In addition, FedMef requires fewer memory footprints than SOTA methods. For example, FedMef significantly reduces the memory footprint of MobileNetV2 by 28.5% while improving the accuracy by more than 2%.

## 2 RELATED WORK

### 2.1 NEURAL NETWORK PRUNING

Neural networks pruning, which emerged in the late 1980s, aims to reduce redundant parameters in deep neural networks (DNNs). Traditional techniques focus on trade-off accuracy and sparsity during inference. This involved ranking parameter importance in a pre-trained DNN and dis-

carding those with lower scores. Various methods determine these scores, such as weight magnitude (Janowsky, 1989; Han et al., 2015) and Taylor expansion of loss functions (Mozer & Smolensky, 1988; LeCun et al., 1989; Molchanov et al., 2019a). A major drawback of these methods is the need to train a dense model first, which increases both computational and memory costs.

Modern pruning research has shifted its focus towards enhancing the efficiency of DNN training processes. For example, dynamic sparse training (Mocanu et al., 2018; Dettmers & Zettlemoyer, 2019; Evci et al., 2020), actively adjusts the architecture of the pruned model throughout the training while maintaining desired sparsity levels. Nevertheless, these methods simply prune low-magnitude parameters and do not address the memory consumption of the activation caches, resulting in decreased accuracy and sub-optimal memory optimization.

## 2.2 Neural Network Pruning in Federated Learning

Federated learning has recently emerged as a promising technique to navigate data privacy challenges in collaborative machine learning (McMahan et al., 2017). However, numerous previous federated pruning efforts (Shao et al., 2019; Li et al., 2021a; Liu et al., 2021; Munir et al., 2021) have encountered setbacks because they rely on the training of dense models on devices, which requires high computation and memory. Thus, they are not suitable for the cross-device FL paradigm, where clients are edge devices with resource constraints.

Recent studies (Qiu et al., 2022; Jiang et al., 2022; Bibikar et al., 2022; Huang et al., 2022) introduce on-device pruning via the dynamic sparse training technique (Mocanu et al., 2018; Dettmers & Zettlemoyer, 2019; Evci et al., 2020). For example, ZeroFL (Qiu et al., 2022) divides the weights into active and nonactive weights for inference and sparsified weights and activation for backward propagation. FedDST (Bibikar et al., 2022) and FedTiny (Huang et al., 2022), inspired by RigL (Evci et al., 2020), perform pruning and growing on devices, with the server generating a new global model through sparse aggregation. However, these methods are unable to reduce the memory footprints of the activation cache, and suffer from significant accuracy degradation after pruning, since they directly prune parameters that may contain important information.

Therefore, all existing federated neural network pruning works fail in creating a specialized pruned model that concurrently satisfies accuracy and memory constraints. Our proposed solution, FedMef, can address all of these issues.

## 2.3 Activation Cache Compression

High-resolution activation tensors are a primary memory burden for modern deep neural networks. Gradient checkpoint (Chen et al., 2016; Gruslys et al., 2016; Feng & Huang, 2021), which stores specific layer tensors and recalculates others during the backward pass, offers a memory saving solution, but at a high computational cost. Alternatively, adaptive precision quantization methods (Chen et al., 2021; Liu et al., 2022; Wang et al., 2023) compress activation caches through quantization but introduce time overhead from dynamic bit width adjustments and dequantization. The activation pruning (sparsification) method (Chen et al., 2022b), which sparsifies activation caches, is lighter than other methods, but relies heavily on batch normalization (BN) layers to guarantee that most of the elements in activation are zero or near zero. Relying on BN layers would be problematic to train with small batches and non-independent and identically distributed (non-i.i.d.) data (Li et al., 2021b; Zhuang & Lyu, 2023). As a result, current activation pruning methods are unsuitable for resource-constrained devices in FL. To address these challenges, our proposed FedMef utilizes scaled activation pruning, effectively compressing activation caches without relying on BN layers.

## 3 Methodology

This section first introduces the problem setup and then outlines the design principles of our proposed FedMef. We then introduce two key components in FedMef: budget-aware extrusion and scaled activation pruning.

Figure 2: The illustration of training pipeline in baseline and the proposed scaled activation pruning method. During the forward pass, the scaled activation pruning generates near-zero activation via the Normalized Sparse Convolution (NSConv). Then, the dense activation caches are pruned based on magnitude. During the backward pass, these pruned caches are used to compute the gradients.

## 3.1 PROBLEM SETUP

In the cross-device FL scenario, numerous resource-constrained devices collaboratively train better models without direct data sharing Kairouz et al. (2021). In this setting, $K$ devices, each with memory and computational constraints, cooperate to train the model with parameters $\theta$. Every device possesses a distinct local dataset, denoted as $\mathcal{D}_k, k \in \{1, 2, \ldots, K\}$. The structure of the pruned model is represented using a mask, $m \in \{0, 1\}^{|\theta|}$, and $\theta \odot m$ denotes the pruned model parameters. Our objective is to derive a specialized sparse model with mask $m$, using the local dataset $\mathcal{D}_k$, to optimize prediction accuracy in FL. During training, the sparsity levels of the mask $s_m$ and the activation caches $s_a$ must be higher than target sparsity ($s_{tm}$ and $s_{ta}$), which is determined by the memory constraints of the devices. Thus, our optimization challenge is to solve the following:

$$\min_{\boldsymbol{\theta}, m} L(\theta, m) := \sum_{k=1}^{K} p_k L_k(\theta, m, \mathcal{D}_k) \quad \text{s.t.} \quad s_m \geq s_{tm}, s_a \geq s_{ta}, \tag{1}$$

where $L_k$ is the loss function of the $k$-th device (*e.g.*, cross-entropy loss), and $p_k$ represents the weight of $k$-th device during model aggregation in the server. Before communicating with the server, each device trains its local model for $E$ local epochs.

## 3.2 DESIGN PRINCIPLES

To ascertain that specialized sparse models can be developed on resource-constrained devices while maintaining privacy, the prevailing trend is to leverage federated dynamic pruning. However, contemporary methods (Jiang et al., 2022; Qiu et al., 2022; Bibikar et al., 2022; Huang et al., 2022) face two pressing issues: significant post-pruning accuracy degradation and high activation memory usage. As illustrated in Figure 1, our framework, FedMef, introduces two solutions to these challenges: budget-aware extrusion and scaled activation pruning. In the FedMef framework, the server starts by distributing a randomly pruned model to devices. Then these devices collaboratively train the sparse models using scaled activation pruning. In this phase, the activation cache is pruned during the forward pass, effectively optimizing memory utilization. After several iterative training rounds, devices employ the budget-aware extrusion technique to transfer vital information from low-magnitude parameters to others. Subsequently, the server adjusts the model structure through magnitude pruning and gradient-magnitude-based growing. Due to the information transfer facilitated by budget-aware extrusion, the post-pruning accuracy degradation is slight. Finally, the framework continues with the training and adjustment of the sparse model until convergence.

To mitigate post-pruning accuracy loss, budget-aware extrusion transfers information from parameters marked for pruning to others, reducing information loss during pruning. Devices achieve this by employing a surrogate loss function with $L_1$ regularization of the low-magnitude parameters. This process not only suppresses their magnitude, but also transfers their information to alternate parameters. Additionally, the devices set up a budget-aware schedule to speed up the extrusion. Subsequently, the server grows and prunes the parameters to create a new model structure. Iterative adjustments ensure the progressive convergence of the model structure to an ideal configuration.

To reduce the memory footprint of the activation caches, we propose scaled activation pruning. Following each layer's forward pass, the activation caches are pruned to reduce memory. During the backward pass, the pruned activation caches are used directly. To ensure that the pruned elements are zero or nearly zero, even when training with a small batch size, we adopt the Normalized Sparse Convolution (NSConv) to reparameterize the convolution layers instead of using batch normalization

layers. Next, we delve into in-depth discussions of budget-aware extrusion and scaled activation pruning techniques.

## 3.3 BUDGET-AWARE EXTRUSION

It is essential to address the information loss that occurs during pruning, as the parameters to be pruned often retain valuable information. Direct pruning can cause a substantial accuracy drop, demanding considerable resources for recovery, as illustrated in Figure 1 (Top right). This issue may become even more pronounced in federated contexts due to the heterogeneous data distribution across devices, potentially amplifying the negative impact on model performance during training.

To address this challenge, we take inspiration from the Dual Lottery Ticket Hypothesis (DLTH) (Bai et al., 2022). The DLTH suggests that a randomly selected subnetwork can be transformed into one that achieves better, or at least comparable, performance to benchmarks. Building on this premise, we introduce budget-aware extrusion within our FedMef framework, which can extrude the information from the parameters to be pruned to other surviving parameters. After sufficient extrusion, the parameters designated for pruning retain only marginal influence on the network, ensuring minimal information and accuracy degradation during the pruning process.

In alignment with the findings of the DLTH (Bai et al., 2022), we employ a regularization term to execute this information extrusion. Given the parameters $\theta$ and its associated mask $m$, the extrusion process on the $k$-th device can be realized through the surrogate loss function $L_k^s$:

$$L_k^s = L_k(\theta, m, \mathcal{D}_k) + \lambda ||\theta_{low}||_2^2, \tag{2}$$

where $\lambda$ is constant and $\theta_{low}$ represents the parameters earmarked for pruning, which is the subset of unpruned parameters $\theta \cdot m$ with the lowest weight magnitudes.

The inherent constraints associated with edge device training resources require that information extrusion should be executed within a limited budget before the pruning process. However, adhering to the original learning schedule represented by $\eta$ is sub-optimal, as in the later epochs of training, the learning rate following traditional decay mechanisms becomes significantly small, impeding the information extrusion process.

To address this issue, we introduce a budget-aware schedule in the context of budget-aware extrusion. The schedule is constructed to accelerate the extrusion process, especially when the original learning rate is insufficient for rapid extrusion. Given $T_{budget}$ as the training budget and $t$ as the present step, the budget-aware learning rate $\beta_t$ is mathematically defined as:

$$\beta_t = p(t)(2\sigma(||\theta_{low}||) - 1)\eta_0, \tag{3}$$

where $\sigma$ is the sigmoid function, $\eta_0$ represents the initial learning rate as per the original learning schedule, and $p(t)$ is the REX schedule factor (Chen et al., 2022a), defined by $p(t) = \frac{2T_{budget} - 2t}{2T_{budget} - t}$. The main objective behind introducing this factor is to effectively adjust the learning rate based on the relative progression of the training and the preset training budget. During the information extrusion process, the learning rate $\mu_t$ is formulated as follows:

$$\mu_t = \max(\eta_t, \beta_t), \tag{4}$$

where $\eta_t$ is the learning rate in the original schedule. This ensures efficient and timely information extrusion by adjusting an adequate learning rate even in the later stages of training. During the normal training stage, the learning rate is $\mu_t = \eta_t$.

In particular, upon receiving the pruned model from the server, the devices mark the parameters $\theta_{low}$ that have the lowest weight magnitude. Then, the devices perform several epochs of budget-aware extrusion with the surrogate loss $L_k^s$ in Equation 2. The learning rate for this phase is dynamic and is governed by the function presented in Equation 4.

After the extrusion phase, each device calculates the Top-K gradients across all parameters and uploads the gradients along with the parameters to the server. The server then aggregates the sparse parameters and gradients to obtain the average parameters and average gradients. Finally, the server prunes the marked $\theta_{low}$ and grows the same number of parameters with the largest averaged gradient magnitude.

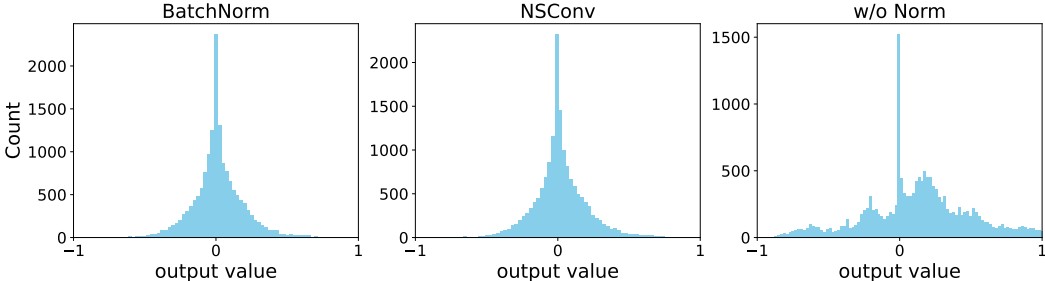

Figure 3: Distribution of output from a convolution layer in ResNet18 using batch normalization layers (BatchNorm), without normalization layers (w/o Norm), and with our proposed Normalized Sparse Convolution (NSConv). The output experiences an internal covariate shift when training without normalization layers, whereas NSConv effectively mitigates this issue. Figure 7 in the appendix shows the output distribution for all convolution layers in the ResNet18 model.

According to the pruning and growing process, the server creates a global model with a new structure, and then FedMef begins to train the new global model. FedMef periodically performs adjustments and training to deliver an optimal sparse neural network suitable for all devices. The detailed algorithm can be viewed in Appendix A.

### 3.4 SCALED ACTIVATION PRUNING

In cross-device FL, where devices may have extremely limited memory, small batch sizes are often employed during training. This diminishes the effectiveness of batch normalization (BN) layers in such a scenario. However, current activation cache compression techniques, such as DropIT (Chen et al., 2022b), are limited in their ability to conduct training without BN layers. To address this issue, we propose a scaled activation pruning technique that achieves superior performance even with small batch sizes.

Given a CNN model with ReLU-Conv ordering, in the $l$-th convolution layer, the sparse filters are represented as $\theta^l \in \mathrm{R}^{ks \times ks \times c_{in} \times c_{out}}$, where $ks$ denotes the kernel size; $c_{in}$ and $c_{out}$ denote the number of input and output channels, respectively. For an input value $a^{l-1}$, the convolution operation in the $l$-th layer that yields the output value $a^l$ is:

$$a^l = \mathrm{Conv}(\theta^l, f(a^{l-1})), \tag{5}$$

where $f(\cdot)$ is any activation function such as ReLU Agarap (2018). Note that $a^{l-1}$ is not only an input of $l$ layer but also the output of the $l-1$-th layer. During the forward pass, the activation $f(a^{l-1})$ must be retained in memory to compute the gradients of the filters $\theta^l$ during the backward pass. Similarly, for each layer, the activation $f(a^l)$ must be stored for later usage, which causes substantial memory footprints.

The activation pruning approach, DropIT (Chen et al., 2022b), prunes $f(a^l)$ in the forward pass. It then uses pruned activation $P(f(a^l))$ for gradient computation in the backward pass. This approach requires that the input $a^l$ be centered around zero. This centering ensures a minimized disparity between the pruned activation $P(f(a^l))$ and its original counterpart $f(a^l)$. However, this zero-centered requirement becomes unattainable when the efficacy of the batch normalization layer decreases. This ineffectiveness arises from internal covariate shift issues (Ioffe & Szegedy, 2015; Brock et al., 2021). Figure 3 shows the mean shift in activation within a ResNet18 model without a normalization layer, resulting in a non-zero mean in the activation distribution. The mathematical details of this effect can be found in Appendix B.1.

To reduce the disparity between the original and pruned activation, inspired by recent methods that remove BN layers (Brock et al., 2021; Zhuang & Lyu, 2023), we introduce Normalized Sparse Convolution (NSConv) into activation pruning. Our primary objective is to ensure that the output of the convolution layer is consistently centered around zero, *i.e.,* the mean value is zero. The convolution operation of NSConv at the $l$-th layer is given by:

$$a^l = \mathrm{Conv}(\hat{\theta}^l, f(a^{l-1})), \tag{6}$$

where $\hat{\theta}^l$ represents the sparse normalized filters with filter-wise weight standardization. The filter-wise standardization formula of the $i$-th sparse filter, denoted by $\hat{\theta}_i^l \in \mathbb{R}^{ks \times ks \times c_{in}}$, is given by:

$$\hat{\theta}_i^l = \gamma \sqrt{c_{in}} \frac{\theta_i^l \odot m_i^l - \mu_\theta}{\sigma_\theta}, \tag{7}$$

where $\theta_i^l \in \mathbb{R}^{ks \times ks \times c_{in}}$ specifies the $i$-th filter of the original filters, $\gamma$ is a constant, and $m_i^l$ denotes the corresponding mask for the sparse filter $\theta_i^l$. The terms $\mu_\theta$ and $\sigma_\theta$ represent the mean and standardization value of the sparse filter $\theta_i^l$, excluding the pruned parameters whose corresponding mask is 0.

**Theorem 1** *Given a CNN model structured in a ReLU-Conv sequence, and allowing the $l$-th convolution layer to perform operations as depicted by the forward pass in Equation 6 and NSConv in Equation 7. For the $i$-th channel of the activation value, $f(a_i^{l-1})$, with its mean and variance denoted as $\mu_f, \sigma_f^2$. The mean and variance for the $i$-th of the output value, $a_i^l$, will be:*

$$\mathbb{E}[a_i^l] = 0, \quad \mathrm{Var}[a_i^l] = \gamma^2(\sigma_f^2 + \mu_f^2). \tag{8}$$

Theorem 1 reveals insights into the capabilities of scaled activation pruning. Specifically, it highlights its efficacy in addressing the disparity between pruned and original activation in CNNs without BN layers. A key factor in its effectiveness is NSConv's ability to normalize the output of each convolution layer, centering it around zero, as shown in Figure 2. By adjusting the hyperparameter $\gamma$, we can control the variance of the distribution, causing a large portion of the activation elements to be either zero or close to it. The proof of Theorem 1 can be found in Appendix B.2.

Incorporating NSConv into scaled activation pruning brings several additional advantages: First, NSConv ignores pruned parameters, focusing solely on the remaining ones. This translates to minimal computational overhead and maintains the sparsity of the normalized parameters. Second, NSConv is suitable for training with small batch sizes because there are no interdependencies between batch elements. Lastly, NSConv ensures uniformity between the training and testing phases.

## 4 EVALUATION

In this section, we dive into an in-depth evaluation of our novel framework, FedMef. We compare it against SOTA frameworks, demonstrating its effectiveness in various testing conditions. Additionally, an ablation study reveals the essential components that make our proposed framework effective.

### 4.1 EXPERIMENTAL SETUP

We assess the effectiveness of FedMef in image recognition tasks by using three datasets: CIFAR-10 (Krizhevsky et al., 2009), CINIC-10 (Darlow et al., 2018), and TinyImageNet (Le & Yang, 2015). We employ the ResNet18 (He et al., 2016) and MobileNetV2 (Sandler et al., 2018) models for evaluation. We conduct experiments on a landscape of 100 devices. The datasets are divided into heterogeneous partitions via a Dirichlet distribution characterized by a factor of $\alpha = 0.5$. We train the models for $R = 500$ federated learning rounds, where each round is composed of $E = 10$ local epochs. We set the training batch size as 64 by default. The target parameters sparsity and target activation sparsity is set to $s_{tm} = 0.9, s_{ta} = 0.9$ by default. The initial learning rate is set as $\eta_0 = 1$ with an exponential decay rate of 0.95. We conducted each experiment five times and reported the average result and standard deviation.

We compare our proposed FedMef with FL-PQSU (Xu et al., 2021), FedDST (Bibikar et al., 2022), and FedTiny (Huang et al., 2022). FL-PQSU is a static pruning method, which employs an initialized pruning based on the $L_1$ norm on the server prior to training. It can be considered as the lower bound of our method. FedDST and FedTiny are state-of-the-art federated dynamic pruning methods. Both of them start with an initial pruned model, subsequently employing mask adjustments to adjust the model architecture The key distinction between them lies in their locus of model structure adjustments: FedTiny centralizes this on the server, whereas FedDST decentralizes it to the devices. Certain federated pruning frameworks, such as ZeroFL (Qiu et al., 2022) and PruneFL (Jiang et al., 2022), which are memory intensive to process dense models, are consciously excluded from our comparison.

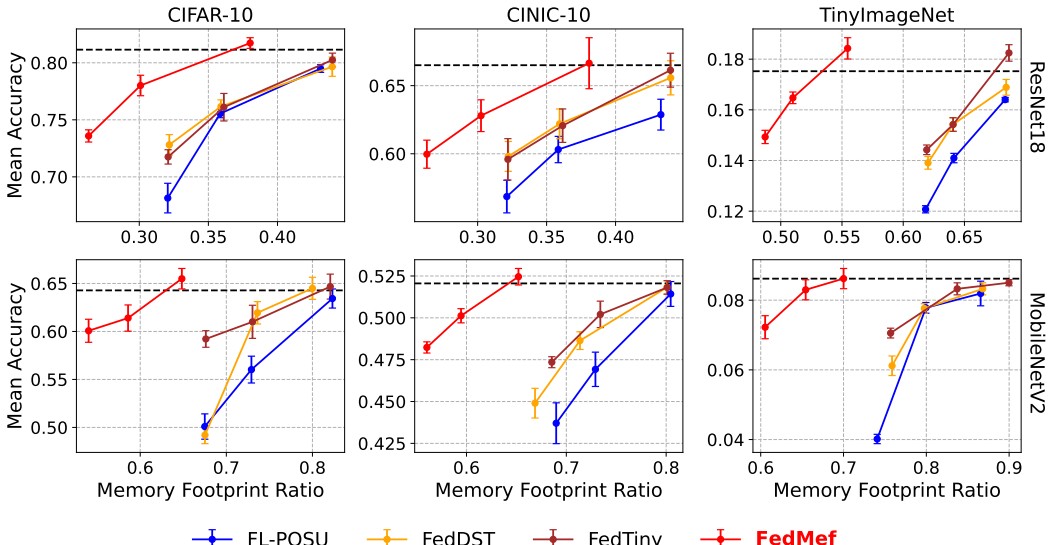

Figure 4: Comparison of accuracy and memory footprint of our proposed FedMef with the existing federated pruning methods on three datasets. The black dashed line marks the accuracy of training a full-size model (without pruning) in FedAvg. The memory footprint ratio is the memory footprint relative to training a full-size model in FedAvg.

In the FedDST, FedTiny and FedMef frameworks, the adjustment of the model structure is applied after $\Delta R = 10$ training rounds. Upon reaching $R_{stop} = 300$ rounds, the framework suspends further adjustment, continuing its training until reaching $R = 500$ rounds. The pruning number for each layer is set to $0.2(1 + cos\frac{t\pi}{R_{stop}E})n$ in the $t$-th iteration, where $n$ is the number of unpruned parameters in the $l$-th layer. Owing to FedDST (Bibikar et al., 2022) necessitating a series of on-device training epochs for fine tuning after adjustment, after 5 epochs of local training, we let FedDST adjust the model structure and then proceed with 5 training epochs. FedTiny's (Huang et al., 2022) adaptive batch normalization module is amputated from our experiments, as its memory overhead renders it infeasible for our device constraints.

## 4.2 PERFORMANCE EVALUATION

To demonstrate the effectiveness of FedMef, we compared it with other frameworks on the CIFAR-10, CINIC-10, and TinyImageNet datasets using ResNet18 and MobileNetV2. A holistic comparison is illustrated in Figure 4. The target sparsity of the parameters is set to $s_{tm} \in \{0.05, 0.1, 0.2\}$. Remarkably, FedMef outperforms all baseline frameworks, both in terms of accuracy and memory efficiency. As a case in point, FedMef achieves an accuracy improvement of $2.13\%$ on the CIFAR-10 dataset with the MobileNetV2 model, while saving $28.5\%$ memory usage compared to the best baseline framework, FedTiny. Such advances in accuracy can be attributed to our proposed budget-aware extrusion, while scaled activation pruning primarily augments memory conservation.

An obvious trend is the superior accuracy benchmarks set by ResNet18 over MobileNetV2 in all datasets. The design of MobileNetV2 is tailored to large-scale image classification, which may make it less suitable for datasets that are relatively small. A noteworthy observation is that FedTiny generally outperforms other baseline methods within comparable memory footprints. Given this empirical trend, ResNet18 is chosen as the default model and FedTiny serves as the primary reference for subsequent experiments.

**Training with Small Batch Size.** Under strict memory constraints, training requires a smaller batch size. However, this compromises statistical robustness and often hinders the effectiveness of batch normalization. To address this issue, we propose scaled activation pruning. The evaluations conducted on the CIFAR-10 dataset with ResNet18, where the batch size is set to 1, as shown in Figure 5 (left), demonstrate that FedMef outperforms all baseline methodologies. The significant improvement in accuracy is mainly due to the use of Normalized Sparse convolution in scaled activation pruning, which is further demonstrated in the appendix B.3.

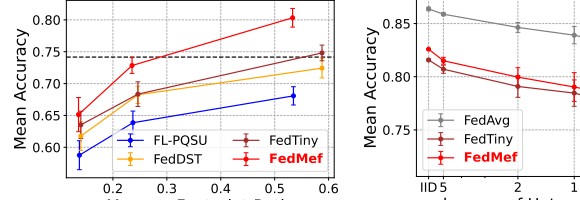 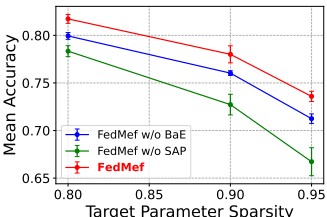

Figure 5: FedMef's average accuracy and standard deviation are compared against: *(left)* various federated pruning frameworks when the training batch size is 1, where the black dashed line represent the accuracy of FedAvg framework; *(middle)* FedAvg and FedTiny across varying degrees of data heterogeneity; *(right)* modified versions of FedMef - one excluding BaE (similar to FedTiny's approach) and the other without SAP (omitting NSConv).

**The impact of adjustment period.** After model structure adjustment, it is necessary to restore accuracy loss through several training rounds. Therefore, the adjustment period should be longer. Unfortunately, given the computational constraints of certain devices, there is an urgent need to limit the number of interval training rounds and local epochs. Empirical results from experiments on the CIFAR-10 dataset with various adjustment periods, $\Delta R$, and a single local epoch are shown in Table 1. When FedTiny performance decreases under resource constraints, FedMef remarkably maintains the performance.

| $\Delta R$ | FedTiny | FedMef |
|---|---|---|
| 3 | 55.09%(1.82%) | 61.94%(0.49%) |
| 5 | 58.73%(1.62%) | 62.12%(0.58%) |
| 10 | 61.18%(1.08%) | 62.77%(0.78%) |

Table 1: Mean accuracy (standard deviation) for FedMef and FedTiny on the CIFAR-10 dataset with various adjustment periods, $\Delta R$.

**Analysis on Different Degrees of Heterogeneity.** We tested the effectiveness of FedMef on heterogeneous data distributions by modulating the Dirichlet distribution's $\alpha$ factor, where lower $\alpha$ indicates a higher degree of heterogeneity. For reference, we compare our results against the full-size model and FedTiny in the CIFAR-10 dataset and the results are shown in Figure 5 (middle). FedMef retains its superior performance compared to the best baseline framework.

## 4.3 ABLATION STUDY

We further analyze the individual contributions of budget-aware extrusion (BaE) and scaled activation pruning (SAP) using trials on the CIFAR-10 dataset with ResNet18. The variants include a FedMef without budget-aware extrusion (akin to FedTiny's mechanism) and a FedMef without scaled activation pruning (mirroring DropIT's approach Chen et al. (2022b) without NSConv). The findings presented in Figure 5 (right) indicate that both budget-aware extrusion and scaled activation pruning boost FedMef's performance. In particular, removing scaling in activation pruning results in substantial information loss during backpropagation and results in performance degradation.

## 5 CONCLUSION

This paper introduces FedMef, a memory-efficient federated dynamic pruning framework designed to generate specialized models on resource-constrained devices in cross-device FL. FedMef addresses the issues of post-pruning accuracy degradation and high activation memory usage that current federated pruning methods suffer from. It proposes two new components: budget-aware extrusion and scaled activation pruning. Budget-aware extrusion reduces information loss in pruning by extruding information from parameters marked for pruning to other parameters within a limited budget. Scaled activation pruning allows activation caches to be pruned to save more memory footprints without compromising accuracy. Experimental results demonstrate that FedMef outperforms existing approaches in terms of accuracy and memory footprint. FedMef reduces the memory footprint by 28.5% compared to the most state-of-the-art method while improving the accuracy by more than 2%

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

# A    ALGORITHM

---

**Algorithm 1** FedMef

---

**Input**: dense initialized parameters $\theta$, $K$ devices with local dataset $\mathcal{D}_1, \dots \mathcal{D}_K$, iteration number $t$, learning rate schedule $\alpha_t$, pruning number $\xi_t^l$ for each layer $l$, the number of local iterations per round $E$, the number of rounds between two adjustment $\Delta R$, and the rounds at which to stop adjustment $R_{stop}$.

**Output**: a well-trained model with sparse $\theta_t$ and specified mask $m_t$

1: $t \leftarrow 0$
2: $\theta_0, m_0 \leftarrow$ random prune dense initialized parameters $\theta$
3: **while** until converge **do**
4:    **for** each device $k = 1$ to $K$ **do**
5:        Fetch sparse parameters $\theta_t$ and mask $m_t$ from the server
6:        **for** $i = 0$ to $E - 1$ **do**
7:            $\hat{\theta}_{t+i}^k \leftarrow$ Filter-wise Sparse Standardization as in Equation 7.
8:            **if** $t \mod \Delta RE = 0$ and $t \leq ER_{stop}$ **then**
9:                Calculate budget-aware learning rate $\beta_{t+i}$ as in Equation 4
10:                $\mu_{t+i} \leftarrow \max(\eta_{t+i}, \beta_{t+i})$
11:                $\theta_{t+i+1}^k \leftarrow \theta_{t+i}^k - \mu_{t+i} \nabla L_k^s(\hat{\theta}_{t+i}^k, m_t, \mathcal{D}_{t+i}^k) \odot m_t$, using scaled activation pruning
12:            **else**
13:                $\theta_{t+i+1}^k \leftarrow \theta_{t+i}^k - \eta_{t+i} \nabla L_k(\hat{\theta}_{t+i}^k, m_t, \mathcal{D}_{t+i}^k) \odot m_t$, using scaled activation pruning
14:            **end if**
15:        **end for**
16:        Upload $\theta_{t+E}^k$ to the server
17:        **if** $t \mod \Delta RE = 0$ and $t \leq ER_{stop}$ **then**
18:            **for** each layer $l$ in model **do**
19:                Compute top-$\xi_t^l$ gradients $\tilde{\boldsymbol{g}}_t^{k,l}$ for pruned parameters with a memory space of $O(a_t^l)$
20:                Upload $\tilde{\boldsymbol{g}}_t^{k,l}$ to the server
21:            **end for**
22:        **end if**
23:    **end for**
24:
25:    The server does
26:    $\theta_{t+E} \leftarrow \sum_{k=1}^{K} \frac{|\mathcal{D}_k|}{\sum_{k=1}^{K} |\mathcal{D}_k|} \theta_{t+E}^k$
27:    **if** $t \mod \Delta RE = 0$ and $t \leq ER_{stop}$ **then**
28:        **for** each layer $l$ in model **do**
29:            $\tilde{\boldsymbol{g}}_t^l \leftarrow \sum_{k=1}^{K} \frac{|\mathcal{D}_k|}{\sum_{k=1}^{K} |\mathcal{D}_k|} \tilde{g}_t^{k,l}$
30:            $\boldsymbol{I}_{grow}^l \leftarrow$ the $\xi_t^l$ pruned indices with the largest absolute value in $\tilde{\boldsymbol{g}}_t^l$
31:            $\boldsymbol{I}_{drop}^l \leftarrow$ the $\xi_t^l$ unpruned indices with smallest weight magnitude in $\theta_{t+E}$
32:            Compute the new mask $m_{t+E}^l$ by adjusting $m_t^l$ based on $\boldsymbol{I}_{grow}^l$ and $\boldsymbol{I}_{drop}^l$
33:        **end for**
34:        $\theta_{t+E} \leftarrow \theta_{t+E} \odot m_{t+E}$ // Prune the model using the updated mask
35:    **else**
36:        $m_{t+E} \leftarrow m_t$
37:    **end if**
38:    $t \leftarrow t + E$
39: **end while**

---

# B    PROOF OF THEOREM 1

In this section, we first introduce the internal covariate shift in CNN without batch normalization layers and then provide the proof of Theorem 1.

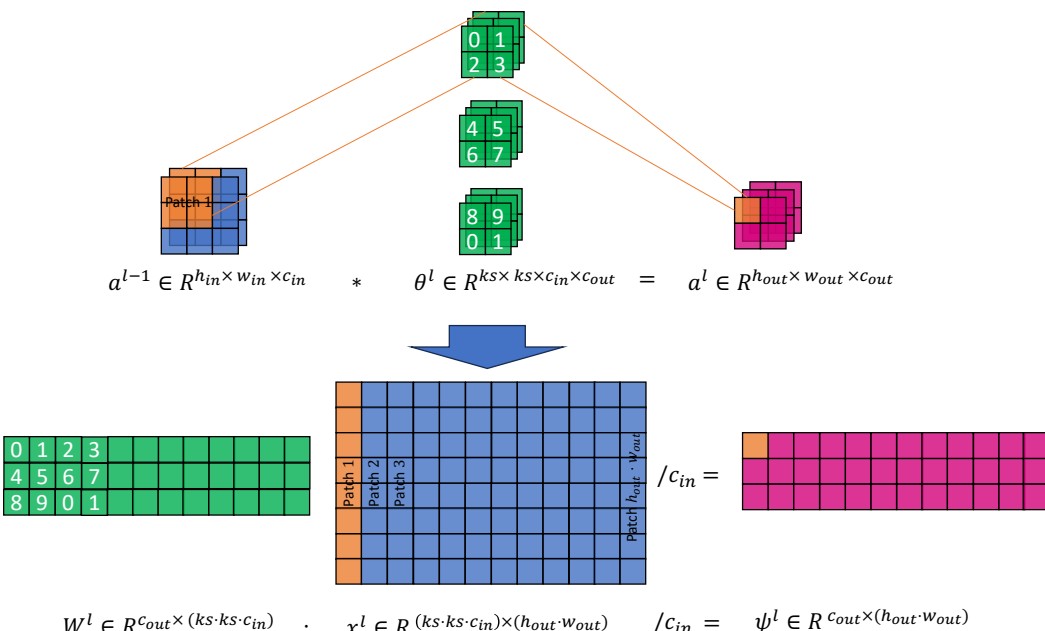

Figure 6: Illustration of transforming the convolution operation into linear multiplication: Begin by flattening each filter from the convolutional filters, $\theta^l$, and stack them to produce the linear weight $W^l$. Next, stack each convolution patch from the input value $a^{l-1}$ to form the linear input $x^l$. The resultant multiplication, $\psi^l$, corresponds to a reshaped version of the original output $a^l$.

## B.1 INTERNAL COVARIATE SHIFT

Given a CNN model with ReLU-Conv ordering, In the $l$-th convolution layer, the sparse filters are represented as $\theta^l \in \mathbb{R}^{ks \times ks \times c_{in} \times c_{out}}$, where $ks$ denotes the kernel size;$c_{in}$ and $c_{out}$ denote the number of input and output channels, respectively. For an input value $a^{l-1} \in \mathbb{R}^{h_{in} \times w_{in} \times c_{in}}$, the convolution operation in the $l$-th layer that yields the output value $a^l \in \mathbb{R}^{h_{out} \times w_{out} \times c_{out}}$ is:

$$a^l = \text{Conv}(\theta^l, f(a^{l-1})), \tag{9}$$

where $f(\cdot)$ is any activation function such as ReLU, leaky ReLU, etc. It is worth noting that $a^{l-1}$ is not just an input; it is also the output of the $l-1$-th layer.

As illustrated in Figure 6, above convolution operation can be converted into a linear multiplicity version as :

$$\psi^l = W^l x^l / c_{in}, \tag{10}$$

where weight matrix $W^l \in \mathbb{R}^{c_{out} \times (ks \cdot ks \cdot c_{in})}$ is the flattening version of the convolution filters $\theta^l$. The $i$-th row of the linear weight $W^l$ is the flattening result of $i$-th filter of original filters, $\theta_i^l$. Linear input $x^l \in \mathbb{R}^{(ks \cdot ks \cdot c_{in}) \times (h_{out} \cdot w_{out})}$ is the stacked convolution patch from the activation $f(a^{l-1})$. The resultant multiplication, $\psi^l$, corresponds to a reshaped version of the original output $a^l$. The $i$-th row of the linear result $\psi^l$ is the flattening result of $i$-th channel of the original output, $a_i^l$.

Denote the mean and variance values of the $i$-th filter of original filters as $\mathbb{E}(\theta_i^l) = \mu_\theta$ and $\text{Var}(\theta_i^l) = \sigma_\theta^2$. Assuming the mean and variance value of the linear input $x^l$ are $\mathbb{E}(x^l) = \mu_x$ and $\text{Var}(x^l) = \sigma_x^2$, the mean and variance of $i$-th channel of output $a_i^l$ will be :

$$\mathbb{E}(a_i^l) = \mathbb{E}(\psi_i^l) = \mathbb{E}(W_i^l)\mathbb{E}(x^l)/c_{in} = \mu_i^\theta \mu_x / c_{in}, \tag{11}$$

$$\text{Var}(a_i^l) = \text{Var}(\psi_i^l) = \text{Var}(W_i^l x^l)/c_{in}^2 = (\sigma_\theta^2 \sigma_x^2 + \sigma_\theta^2 \mu_x^2 + \mu_\theta^2 \sigma_x^2)/c_{in}^2, \tag{12}$$

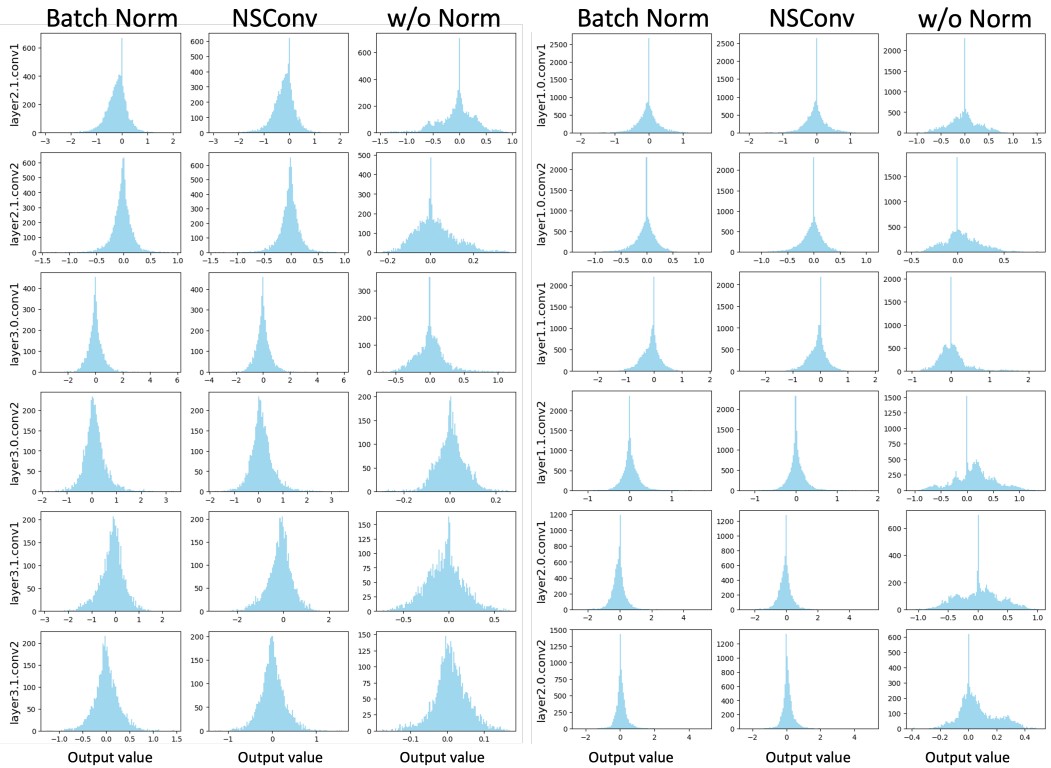

Figure 7: Distribution of output from all convolution layers in ResNet model using Batch Normalization layers (BatchNorm), without normalization layers (w/o Norm), and with Normalized Sparse Convolution (NSConv).

Consider $f(\cdot)$ to be the activation function of ReLU, which implies that the input value $\mu_x$ has a positive mean. During training, the mean value of each filter $\theta_i^l$ is difficult to keep in zero. Therefore, without batch normalization layer, the mean of output from the convolution layer will be unable to reach around zero.

## B.2 PROOF

**Theorem** *Given a CNN model structured in a ReLU-Conv sequence, and allowing the l-th convolution layer to perform operations as depicted by the forward pass in Equation 6 and NSConv in Equation 7. For the $i$-th channel of the activation value, $f(a_i^{l-1})$, with its mean and variance denoted as $\mu_f, \sigma_f^2$. The mean and variance for the $i$-th of the output value, $a_i^l$, will be:*

$$\mathbb{E}[a_i^l] = 0, \quad \text{Var}[a_i^l] = \gamma^2(\sigma_f^2 + \mu_f^2). \tag{13}$$

*Proof.* As illustrated in Figure 6, convolution operation can be converted to a linear multiplicity version as:

$$\psi^l = \hat{W}^l x^l / c_{in}, \tag{14}$$

where weight matrix $\hat{W}^l \in \mathbb{R}^{c_{out} \times (ks \cdot ks \cdot c_{in})}$ is the flattening version of sparse normalized convolution filters $\hat{\theta}^l$. The $i$-th row of the linear sparse weight $\hat{W}_i^l$ is the flattening result of the $i$-th filter of normalized filters, $\hat{\theta}_i^l$.

Therefore, the mean and variance of the $i$-th row of normalized linear weight, $\hat{W}_i^l$ are $\mathbb{E}(\hat{W}_i^l) = 0$ and $\text{Var}(\hat{W}_i^l) = \gamma^2 c_{in}$. the mean and variance for the $i$-th of the output value will be :

$$\mathbb{E}(a_i^l) = \mathbb{E}(\psi_i^l) = \mathbb{E}(\hat{W}_i^l)\mathbb{E}(x^l)/c_{in} = 0, \tag{15}$$

$$\mathrm{Var}(a_i^l) = \mathrm{Var}(\psi_i^l) = \mathrm{Var}(\hat{W}_i^l x^l)/c_{in}^2 = \gamma^2(\sigma_x^2 + \mu_x^2), \tag{16}$$

Because the linear input $x^l$ is the sampled version of the input activation $f(a^{l-1})$, considering the randomness, the mean and variance of the linear input $x^l$ will be $\mu_x = \mu_f$, $\sigma_x^2 = \sigma_f^2$. Therefore, we can get:

$$\mathbb{E}(a_i^l) = 0, \quad \mathrm{Var}(a_i^l) = \gamma^2(\sigma_f^2 + \mu_f^2). \tag{17}$$

### B.3 EXPERIMENT RESULT

To assess the effectiveness of our proposed Normalized Sparse Convolution (NSConv), we conducted experiments on the CIFAR-10 dataset with the ResNet18 model in our proposed FedMef framework, with the sparsity of target parameters set to 0.9. The results of the experiment, shown in Figure 7, demonstrate that NSConv can achieve an effect similar to that of a Batch Normalization layer. Additionally, the activation values of ResNet18 without normalization decrease and the distribution becomes more centralized as the layer deepens, further supporting Equations 12 and 12, which indicate that the mean and variance values will be scaled with $1/c_{in}$ and $1/c_{in}^2$, respectively.

## C   CALCULATING TRAINING MEMORY FOR MODELS

### C.1   COMPRESSION SCHEMES

The storage for a matrix consists of two components, values and positions. The aim of compression is to reduce the storage of the positions of non-zero values in the matrix. Suppose we want to store the positions of $m$ non-zeros value with $b$ bit-width in a sparse matrix $M$. The matrix $M$ has $n$ elements and a $n_r \times n_c$ shape. Depending on the density $d = m/n$, we apply different schemes to represent the matrix $M$. We use $o$ bits to represent the positions of $m$ non-zero values and denote the overall storage as $s$.

- For density $d \in [0.9, 1]$, **dense** scheme is applied, i.e. $s = n \cdot b$.
- For density $d \in [0.3, 0.9)$, **bitmap** (BM) is applied, which stores a map with $n$ bits, i.e. $o = n, s = o + mb$.
- For density $d \in [0.1, 0.3)$, we apply **coordinate offset** (COO), which stores elements with its absolute offset and it requires $o = m\lceil \log_2 n \rceil$ extra bits to store position. Therefore, the overall storage is $s = o + mb$
- For density $d \in [0., 0.1)$, we apply **compressed sparse row** (CSR) and **compressed sparse column** (CSC) depending on size. It uses column and row index to store the position of elements and $o = m\lceil \log_2 n_c \rceil + n_r\lceil \log_2 m \rceil$ bits are needed for CSR. The overall storage is $s = o + mb$

For tenor, we carry out reshaping prior to compression. This approach allows us to determine the memory needed to train the network's parameters.

### C.2   THE MEMORY FOOTPRINT OF TRAINING MODELS

We estimate the memory footprint for training to be a combination of parameters, activations, gradients of activations, and gradients of parameters. The memory for parameters is equal to the storage of parameters. We estimate the memory for activations by taking the maximum value of multiple measurements. For simplicity, we set the memory for gradients of activations to be equal to the memory for activations. We do not consider the memory for hyper-parameters and momentum. Assuming the memory for dense and sparse parameters are $M_d^p$ and $M_s^p$ respectively, and the memory for dense and sparse activations is $M_d^a$ and $M_s^p$, the overall training memory for each algorithm would be:

- **FedAvg** These techniques necessitate the training of a dense model, thus the memory for the gradients of parameters is close to $M_d^p$. The memory footprint for training is approximately $2M_d^p + 2M_d^a$.

- **FL-PQSU** These techniques teach a lean static model, so the memory for gradients of parameters is close to $M_s^p$. The memory needed for training is approximately $2M_s^p + 2M_d^a$.

- **FedTiny and FedDST.** Since these methods only update the top-K gradients in memory to adjust the model structure, the extra memory is used to store the top-$\xi_l^t$ gradients and their indices in one block. Therefore, the memory for the parameter gradients is approximately $M_s^p + 3b\sum_l \xi_l^t$, where $b$ is the bit width. Consequently, the overall memory footprint is $2M_s^p + 2M_d^a + 3b\sum_l \xi_l^t$.

- **FedMef.** In comparison to FedTiny and FedDST, FedMef applies scaled activation pruning to activation, resulting in a cache memory of activation of $M_d^a$. However, the activation gradients are not pruned, leading to a total memory footprint of $2M_s^p + M_s^a + M_d^a + 3b\sum_l \xi_l^t$.

