# OpenReview forum: "FedMef: Towards Memory-efficient Federated Dynamic Pruning"
_ICLR.cc/2024/Conference — ICLR 2024 Conference Withdrawn Submission_

### Official Review · Reviewer_vuu4 · 2023-10-31

**Soundness:** 2 fair
**Presentation:** 1 poor
**Contribution:** 2 fair
**Rating:** 3
**Confidence:** 5

**Summary:**

This paper proposed FedMef, a memory-efficient dynamic pruning framework for federated learning environments. FedMef incorporated budget-aware extrusion and scaled activation pruning to guarantee the model pruning performance and memory efficiency. The proposed FedMef shows benefits in memory-limited federated learning edge scenarios.

**Strengths:**

- The activation pruning in FedMef is able to reduce the memory footprint.
- FedMef introduces budget-aware extrusion to compensate for post-pruning performance loss.
- Adequent reference to discuss related works
- Experiments show performance improvement compared to baseline

**Weaknesses:**

1. The presentation should be polished. Although the grammar and fluency of the presentation is good, however:
- there are a lot of redundant/overlapped sentences paragraph throughout the introduction, methodology, and experiments, which raises unclear and confusion
- Figure 1 needs further polished. It shows confusion and unclear, reader can not catch information from it

2. Limited experiments. The experiments cannot support the arguments the paper proposes.

3. For the federated learning optimization results, the authors only put the final model average performance after training is finished.   However, the training stableness is also an important indicator to evaluate the performance of an FL algorithm. The model-performance vs. #communication round figures should be present in the paper. Especially, pruning within training possibly introduces unstable training factors.

 4.  The paper states their introduced budget-aware extrusion can maintain pruning efficiency while preserving post-pruning performance, which is a key contribution of this paper. However, there are no experiments that direct point to this argument. This is a key points that stands FedRef from other baselines, and readers expect to see more persuasive experiments on it.

5. The proposed pruning method introduces extra communication overhead, such as the transfer of the masking. Evaluation on communication cost and comparison to baselines is also important.


6. Limited model architecture to be tested. To show the proposed method is universal, more neural architecture can be evaluated.

**Questions:**

Please address the points I proposed in the weakness section, besides that:
1. Improve the presentation style,
2. Improve the evaluation. The key contributions of this paper stand out in other dynamic pruning FL solutions is preserving post-pruning performance and reducing the activation memory.
Provide a comprehensive analysis of the results, highlighting the key contributions of your work in dynamic pruning FL solutions.
Discuss how your approach preserves post-pruning performance and reduces activation memory in detail in the experiments using tables, figures etc. If possible, provide additional experimental results or data to counteract the weaknesses.

---

### Official Review · Reviewer_pgar · 2023-11-01

**Soundness:** 4 excellent
**Presentation:** 4 excellent
**Contribution:** 3 good
**Rating:** 6
**Confidence:** 4

**Summary:**

The paper aims to reduce memory consumption when deploying FL to resource-constrained devices. To achieve this goal, the authors proposed FedMef, a memory-efficient FL training framework with two key components: budget-aware extrusion to maintain pruning efficiency and scaled activation pruning to reduce memory usage. The authors have conducted quite extensive experiments which show that the proposed method effectively reduces memory footprint.

**Strengths:**

+ The problem this paper focuses on is important and timely for cross-device FL.
+ The shining point of the work is the two newly proposed components that reduce the activation memory while maintaining or even achieving better performance than the compared counterparts.
+ The experiments are quite solid.
+ The paper is well written. The motivation and problem statement are clear and easy to understand.

**Weaknesses:**

- The paper mainly focused on cross-device FL, it would be interesting to discuss whether the proposed method is suitable for cross-silo FL.
- Some sensitivity analysis is missing, for instance, what is the impact of local epochs and the number of clients selected in each round?

**Questions:**

Please refer to above.

---

### Official Review · Reviewer_Lm19 · 2023-11-01

**Soundness:** 2 fair
**Presentation:** 2 fair
**Contribution:** 2 fair
**Rating:** 3
**Confidence:** 4

**Summary:**

FedMeF is a federated learning framework that implements memory-efficient dynamic pruning. This work proposes using a budget-aware extrusion for addressing post-running accuracy and a scaled activation pruning for tackling high activation memory. Experiments show improved performance on various datasets and using different models.

**Strengths:**

+ The experiments show that FedMeF outperforms recent works like FedTiny and FedDST on the CIFAR-10, CINIC-10 and TinyImageNet datasets interns of accuracy given comparable memory footprints.

**Weaknesses:**

- The authors only focused on comparing the pruned models without showing how much communication cost + computation cost was required to achieve such performance. In other words, the cost of pruning, interns of communication and computation cost isn’t discussed. I assume this could be an important metric to consider when comparing against other pruning methods.
- Presentation could be improved. For example, Figure 1 could be improved.
- The experiment done by the author is not enough  and is done with a limited number of architecture

**Questions:**

Please check the Weaknesses Section fro details and address those points.
- The pruning seems architecture-specific. Specifically, it works with convolutional layers since we are replacing BN layers with NSConv. Is this correct? The authors should clarify it.
- Could you consider communication cost and computation cost when comparing against other pruning methods?

---

### Official Review · Reviewer_fyPU · 2023-11-09

**Soundness:** 2 fair
**Presentation:** 2 fair
**Contribution:** 2 fair
**Rating:** 3
**Confidence:** 1

**Summary:**

This paper proposes FedMef, a federated learning framework for training specialized sparse neural network models on resource-constrained edge devices. Federated learning enables collaborative model training across decentralized edge devices without direct data sharing. However, existing federated pruning techniques face challenges of significant accuracy loss after pruning and high activation memory usage. To address these issues, FedMef introduces two key components - budget-aware extrusion and scaled activation pruning. Budget-aware extrusion transfers essential information from low-magnitude parameters marked for pruning to others to minimize information loss. Scaled activation pruning allows pruning activation caches to reduce memory footprint without compromising accuracy. Specifically, it leverages Normalized Sparse Convolution layers for more effective activation pruning even with small batches. Experiments on image classification datasets demonstrate FedMef's superior performance over state-of-the-art methods, reducing memory footprint by 28.5% while boosting accuracy by over 2%. The proposed techniques enable efficient on-device federated learning to develop specialized sparse models that meet accuracy and memory constraints.

**Strengths:**

The paper has the following strengths:

1) The paper addresses important challenges in federated learning - accuracy degradation and high memory usage during on-device training. This makes it highly relevant given the growth of federated learning.
2) The proposed techniques - i) budget-aware extrusion and ii) scaled activation pruning - are interesting and provide effective solutions to the mentioned challenges.
3) The methods are well-motivated through mathematical analysis and derivations. For example, the analysis showing how Normalized Sparse Convolution addresses internal covariate shift.
4) The experiments are extensive, spanning 3 datasets and 2 models. Comparisons to state-of-the-art methods demonstrate clear improvements.
5) The writing is clear and well-structured. The problem context is well-established and the proposed techniques are described in sufficient detail. Tables and figures aid understanding.

**Weaknesses:**

The paper can be improved based on the following points.

1) The computational overhead of the proposed methods is not analyzed. What is the additional computational cost of budget-aware extrusion and scaled activation pruning?
2) The convergence behavior compared to baselines is not mentioned in the paper. Does your method converge faster or slower than the baselines?
3) Only computer vision tasks are evaluated. Similarly, have you explored your method on other tasks like NLP (non-computer vision based tasks)? If not, how do you expect it to perform? Can you provide justification on this topic?
4) The sensitivity to key hyperparameters like the regularization coefficient λ is not investigated. How sensitive is the performance to the choice of λ? Is there a principled way to set it?
5) The communication overhead between clients and server is not compared. How does your method affect the uplink/downlink communication compared to the baselines? Can you provide comparative study?
6) Only model size and activation memory are considered for memory footprint. Have you measured the actual memory consumption during training compared to baselines? This has not been mentioned properly I the paper.
7) The theoretical analysis makes simplifying assumptions about the data distribution. How do you expect your analysis to hold for more complex real-world data distributions? Can you provide justification/analysis on this?

**Questions:**

Based on the points mentioned in the weaknesses section of the paper above, the following questions need to answered in the paper to improve reproducibility and understanding of the proposed work.

1) The computational overhead of the proposed methods is not analyzed. What is the additional computational cost of budget-aware extrusion and scaled activation pruning?
2) The convergence behavior compared to baselines is not mentioned in the paper. Does your method converge faster or slower than the baselines?
3) Only computer vision tasks are evaluated. Similarly, have you explored your method on other tasks like NLP (non-computer vision based tasks)? If not, how do you expect it to perform? Can you provide justification on this topic?
4) The sensitivity to key hyperparameters like the regularization coefficient λ is not investigated. How sensitive is the performance to the choice of λ? Is there a principled way to set it?
5) The communication overhead between clients and server is not compared. How does your method affect the uplink/downlink communication compared to the baselines? Can you provide comparative study?
6) Only model size and activation memory are considered for memory footprint. Have you measured the actual memory consumption during training compared to baselines? This has not been mentioned properly I the paper.
7) The theoretical analysis makes simplifying assumptions about the data distribution. How do you expect your analysis to hold for more complex real-world data distributions? Can you provide justification/analysis on this?